# Structural and Energetic Affinity of Annocatacin B with ND1 Subunit of the Human Mitochondrial Respiratory Complex I as a Potential Inhibitor: An In Silico Comparison Study with the Known Inhibitor Rotenone

**DOI:** 10.3390/polym13111840

**Published:** 2021-06-02

**Authors:** Camilo Febres-Molina, Jorge A. Aguilar-Pineda, Pamela L. Gamero-Begazo, Haruna L. Barazorda-Ccahuana, Diego E. Valencia, Karin J. Vera-López, Gonzalo Davila-Del-Carpio, Badhin Gómez

**Affiliations:** 1Centro de Investigación en Ingeniería Molecular—CIIM, Universidad Católica de Santa María, Urb. San José s/n—Umacollo, Arequipa 04013, CP, Peru; cfebres@ucsm.edu.pe (C.F.-M.); jaguilar@ucsm.edu.pe (J.A.A.-P.); pgamero@ucsm.edu.pe (P.L.G.-B.); hbarazorda@ucsm.edu.pe (H.L.B.-C.); dvalenciac@ucsm.edu.pe (D.E.V.); 2Laboratory of Genomics and Neurovascular Diseases, Universidad Católica de Santa María, Arequipa 04013, CP, Peru; kvera@ucsm.edu.pe; 3Facultad de Ciencias Farmacéuticas, Bioquímicas y Biotecnológicas, Universidad Católica de Santa María, Urb. San José s/n—Umacollo, Arequipa 04013, CP, Peru; gdavilad@ucsm.edu.pe; 4Vicerrectorado de Investigación, Universidad Católica de Santa María, Arequipa 04013, CP, Peru

**Keywords:** annocatacin B, ND1 subunit, mitochondrial respiratory complex I, MRC-I, molecular dynamics simulations, MD, Hirshfeld charges, MM/PBSA

## Abstract

ND1 subunit possesses the majority of the inhibitor binding domain of the human mitochondrial respiratory complex I. This is an attractive target for the search for new inhibitors that seek mitochondrial dysfunction. It is known, from in vitro experiments, that some metabolites from *Annona muricata* called acetogenins have important biological activities, such as anticancer, antiparasitic, and insecticide. Previous studies propose an inhibitory activity of bovine mitochondrial respiratory complex I by bis-tetrahydrofurans acetogenins such as annocatacin B, however, there are few studies on its inhibitory effect on human mitochondrial respiratory complex I. In this work, we evaluate the in silico molecular and energetic affinity of the annocatacin B molecule with the human ND1 subunit in order to elucidate its potential capacity to be a good inhibitor of this subunit. For this purpose, quantum mechanical optimizations, molecular dynamics simulations and the molecular mechanics/Poisson–Boltzmann surface area (MM/PBSA) analysis were performed. As a control to compare our outcomes, the molecule rotenone, which is a known mitochondrial respiratory complex I inhibitor, was chosen. Our results show that annocatacin B has a greater affinity for the ND1 structure, its size and folding were probably the main characteristics that contributed to stabilize the molecular complex. Furthermore, the MM/PBSA calculations showed a 35% stronger binding free energy compared to the rotenone complex. Detailed analysis of the binding free energy shows that the aliphatic chains of annocatacin B play a key role in molecular coupling by distributing favorable interactions throughout the major part of the ND1 structure. These results are consistent with experimental studies that mention that acetogenins may be good inhibitors of the mitochondrial respiratory complex I.

## 1. Introduction

It has been almost 100 years since Warburg presented the first connection between the mitochondria and tumors appearance [1]. The mitochondria fulfill an energetic role in cells, specifically in cancer cells; this role is essential for developing tumors through glycolysis [2,3]. On that basis, several mechanisms associated with tumor generation, such as loss of enzymatic function, mitochondrial genome mutation, reprogramming of mitochondrial metabolism, have been studied [4,5]. Although controversial [6], some hypotheses and studies show that, to a greater or lesser extent, neoplastic cells have many phenotypes related to their energy production, from high aerobic glycolysis, through a partially active oxidative phosphorylation, to a highly productive one [7,8].

For instance, the mitochondrial respiratory complex I (MRC-I) is directly involved in the appearance of colorectal cancer [9], prostate cancer [10], endometrial cancer [11], breast cancer [12], and melanoma [13]. Thus, this complex protein has become a therapeutic target to develop anticancer drugs. Besides, the MRC-I catalyze the formation of reactive oxygen species (ROS).

MRC-I, also named ubiquinone oxidoreductase , has a molecular mass of approximately 1 MDa; its structural conformation is composed of fourteen central subunits. ND1 subunit is one of those and has most of the inhibitor binding domain in the ubiquinone oxidoreductase. To date, one the main known inhibitors of the MRC-I is the rotenone molecule [14]. Rotenone is an isoflavone compound and has been found in many *Fabaceae* plants. Furthermore, it was used as a pesticide and piscicide [15] due to its high toxicity [16,17]. Its effect on cancer cell lines has been evaluated in vitro, showing the inhibition of proliferation and induction of apoptosis [18,19]. Nevertheless, its toxicity in cells complicates its use as an anticancer drug, mainly because it is highly neurotoxic due to its lipophilic nature and the fact that it does not need an extra metabolism to be active or transporter to enter neurons [14,20,21]. Consequently, the challenge is to find new inhibitors that could be less toxic than rotenone.

Murai et al. analyzed rotenone and a synthetic acetogenin as an inhibitor of the bovine heart MRC-I [22]. They revealed that acetogenins are involved in the binding domain of several inhibitors as rotenone does. In fact, acetogenins with two adjacent tetrahydrofurans (THF) rings were reported to show higher antitumor activity and toxicity than those that had only one THF [23], and have been found in the family of *Annonaceae*, i.e., soursop (*Annonamuricata*) [24].

In traditional medicine, soursop also has important uses, including anticonvulsant, antiarthritic, antiparasitic, hepatoprotective, etc. Many of these beneficial attributes have been ascribed to acetogenins [24,25]. One of the most studied properties in soursop is its potential anticarcinogenic effect due, in a way, to its powerful cytotoxic features [25,26]. It has been possible to isolate more than 100 acetogenins from different parts of the *Annonaceae* plants [24,25,27]. The effect of acetogenins as inhibitors of the MRC-I has been suggested and demonstrated for more than 20 years [28].

Acetogenins have showed important behaviors when evaluating their potential cytotoxic activity against cancer cells; some of these molecules already have proven anti-cancer properties, such as bullatacin, motrilin, assimin, trilobacin, annonacin, gianttronenin, and squamocin. However, we still do not have enough information about most of the acetogenins [29]. The main characteristic of acetogenins’ molecular structure is their linear 32 to 34 carbon chains containing oxygen-containing functional groups. Annocatacin B is an acetogenin with two adjacent THF rings and has been identified in the leaves of soursop; it has also been reported that it possesses toxicity against human hepatoma cells [24,30]. Currently, there is not much information about annocatacin B; so, it has a great potential for new research. In that sense, the objective of this work was to determine the plausible inhibitory role of annocatacin B with the ND1 subunit compared with rotenone as a control, considering all this as a challenge in the search for new inhibitors of MRC-I. To accomplish this, we applied computational techniques as quantum mechanical (QM) calculations, molecular dynamics (MD) simulations, and molecular mechanics/Poisson–Boltzmann surface area (MM/PBSA) calculations.

## 2. Computational Details

### 2.1. Structural Preparation

We analyzed two molecules as ligands to the ND1 complex, rotenone (PubChem ID 6758) and annocatacin B (PubChem ID 10483312) (Figure 1a). The structures of both molecules were built using the GaussView v.6 software package [31], and optimized by DFT calculations using Gaussian 16 software package [32] (Figure 1b). The optimization process were performed using the CAM-B3LYP exchange-correlation functional [33], and the TZVP basis set [34]. The vibrational frequencies were calculated to ensure that the geometries were those of the minimum energy. In order to investigate the electrostatic effect of the ligands on the ND1 complex, atomic charges were calculated using the Hirshfeld population analysis [35,36,37] with implicit solvent effect (*SCRF = (SMD, Solvent = Water)*), and molecular electrostatic potential (ESP) surfaces were used to visualize the polar and non-polar regions of these ligands. To obtain the MD parameters and topologies of the ligands, we used the TPPMKOP server [38], which uses the parameters of the OPLS-AA force field to generates them [39,40]. These topologies were reparametrized using the optimized structures and atomic charges obtained in previous quantum calculations.

On the other hand, the phospholipid bilayer membrane was built with 512 dipalmit- oyl-phosphatidylcholine (DPPC) molecules. A 128-DPPC bilayer with 64 lipid molecules in each layer was replicated four times (twice in both the *x* and *y* directions), to obtain the membrane model. The InflateGRO methodology was used for the embedding of ND1 protein in the lipid membrane [41].

The three-dimensional crystallographic structure of the human MRC-I was considered for this study and obtained from the Protein Data Bank (PDB) by the PDB ID: 5XTD [42]. Crystallographic water molecules were removed in Chimera UCSF 1.11.2 [43]. From this MRC-I, the structure of the ND1 subunit was extracted, since it largely possesses the quinone-binding domain between residues Y127 and K262 (according to the ND1 subunit nomenclature).

### 2.2. MD Simulations

Molecular dynamics (MD) simulations were carried out in Gromacs 2019 [44] with the OPLS-AA force field. Firstly, we performed an energy minimization of the whole protein in the vacuum with the steepest descent algorithm with a maximum of 50,000 steps. Then, the DPPC parameters for the lipid bilayer were obtained from the work of Peter Tieleman et al. [45]. The new system (protein + DPPC membrane) was located in the center of a cubic box with a 1.0 nm distance between the system-surface and the box edge on *z* axis. SPC water-model molecules and ions were added to neutralize the systems. Next, we proceeded with another energy minimization with a maximum of 50,000 steps. The equilibrium MD simulation was realized with position restraint in two ensembles. The first was the canonical ensemble (NVT) at 323.15 K with a trajectory of 50 ps using a V-rescale thermostat. The second was the isobaric-isothermal ensemble (NPT) at 309.65 K, with semi-isotropic pressure coupling, the compressibility of 4.5 × 10${}^{-5}$, and a reference pressure of 1.0 bar for along the 50 ps of the trajectory using the Nosé–Hoover thermostat and the Parrinello–Rahman barostat. The production of MD without position restrain was calculated in the isobaric-isothermal ensemble at 309.65 K and semi-isotropic pressure coupling (same equilibrium condition of NPT ensemble) for 500 ns of trajectory. Periodic boundary conditions (PBC) in all directions, particle mesh Ewald (PME) algorithm for long-range electrostatics with cubic interpolation with a cut-off of 0.9 nm, and linear constraint solver (LINCS) with all bonds constrained were applied for all MD simulations.

### 2.3. Molecular Docking Calculations

First, the coupling was made between the ND1 subunit and rotenone, and then, between ND1 and annocatacin B. To accomplish this, we used PATCHDOCK server [46,47], a molecular docking algorithm based on shape complementarity principles, and we selected the top score solution for each of the two systems, because these top score structures were in agreement with the experimental data [48]. 4.0Å clustering RMSD and default mode parameters were used. Later, we took these top score solution complexes and introduced them into the lipid bilayer/water systems. Subsequently, we carried out the MD simulations of the systems: ND1—rotenone and ND1—annocatacin B, following the aforementioned steps.

### 2.4. MM/PBSA Calculations

To evaluate the binding affinities of ND1-ligand interactions, we performed the molecular mechanics Poisson–Boltzmann surface area (MM/PBSA) calculations [49]. This was made using the g_mmpbsa program [50], which calculates components of binding energy using the MM/PBSA method except the entropic term using a energy decomposition scheme. Despite g_mmpbsa not including the calculation of entropic terms and therefore not being able to calculate the absolute binding free energies (BFE), as Kumari et al. stated [50], it does calculate the relative BFE. So, we used this tool to compare different ligands that bind to the same receptor protein. Calculations of free energies and energy contributions by residue were carried out in order to localize the main residue interactions and to assess the effect of each residue on the ND1—ligand complexes. The last 200 ns of the MD trajectories were analyzed at a 1 ns time interval to estimate the binding free energy (ΔG${}_{bind}$), which was calculated using the following equation:(1)$$\Delta G_{bind}=G_{complex}-\left(G_{ND1}+G_{lig}\right)=\Delta E_{MM}+\Delta G_{sol}-T\Delta S$$
where $G_{complex}$ is the total free energy of the ND1-ligand complexes; $G_{ND1}$ and $G_{lig}$, are the free energies of isolated ND1 structure and rotenone or annocatacin B in solvent. $\Delta E_{MM}$, represents the molecular mechanics energy contributions; $\Delta G_{sol}$ is the free energy solvation required to transfer a solute from vacuum into the solvent. The TΔS term refers to the entropic contribution and was not included in this calculation due to the computational costs [50,51,52]. Therefore, individual $E_{MM}$, and $G_{sol}$ terms were calculated as follows:(2)$$E_{MM}=E_{bonded}+E_{vdW}+E_{elec}$$
(3)$$G_{sol}=G_{p}+G_{np}=G_{p}+\gamma A$$

In Equation (Equation 2), the bonded interactions are represented by the $E_{bonded}$ term, and in the single-trajectory approach, $\Delta E_{bonded}$ is taken as zero [49]. The non-bonded interactions are represented by the $E_{vdW}$ and $E_{elec}$ terms. In Equation (Equation 3), the solvation free energy of ($G_{sol}$), is the sum of the polar ($G_{p}$) and non polar ($G_{np}$) contributions. The $G_{p}$ term is calculated by solving the Poisson–Boltzmann equation, while for the $G_{np}$ term, we used the SASA nonpolar model, where γ (0.0226778 kJ/mol A2) is a coefficient related to the surface tension of the solvent, and A is SASA value. In order to ensure the convergence of our MM/PBSA results, we have considered only the last stable 200 ns (20 frames) of the MD trajectories and were assessed using the FEL analyses from each complex. The frames were selected at a regular interval of 1 ns for better structure–function correlation. In addition, we used the bootstrap analysis to calculate the average binding energy included in the g_mmpbsa tools. All calculations were obtained at 309.65 K, and default parameters were used to calculate molecular mechanics potential energy and solvation free energy [50]. Finally, the binding free energy by residue was obtained using:(4)$$\Delta G_{bind}^{res}=\Delta E_{MM}^{res}+G_{p}^{res}+G_{np}^{res}$$

### 2.5. Structure and Data Analysis

Statistical results, root mean squared deviation (RMSD), root mean squared fluctuation (RMSF), radii of gyration (RG), solvent accessible surface area (SASA), hydrogen bonds (HB), binding free energies (BFE), matches, structures, trajectories, B-factor maps, were obtained using Gromacs modules. An analysis of structure properties was performed using the MD trajectories of the last 200 ns of each simulations, then visualized using Visual Molecular Dynamics (VMD) software [53] and UCSF Chimera v.1.14 [43]. The graphs were plotted using XMGrace software [54]. Moreover, 2D representations of electrostatic and hydrophobic interactios were built using LigPlot program [55]. The ESP surfaces within the molecular mechanics framework were calculated in APBS (Adaptive Poisson Boltzmann Surface) software v.1.4.1, [56] and the pqr entry was created in the PDB2PQR server [57]. Free Energy Landscape (FEL) maps were used to visualize the energy associated with the protein conformation of the different models during the MD simulations. These maps are usually represented by two variables related to atomic position and one energetic variable, typically Gibbs free energy. In this work, we considered two substructures of ND1 protein for the FEL map analysis, Site A (Y127 to F198) and Site B (D199 to K262). These two regions were adopted from the work of Kakutani et al. [48]. The FEL maps were plotted using *gmx sham* module, while the RMSD and RG were considered as the atomic position variables with respect to its average structure and figures were constructed using Wolfram Mathematica 12.1 [58].

## 3. Results and Discussion

The human MRC-I belongs to a highly organized supercomplex, named respirasome. The complexes I, III, and IV arise in a more stable fashion at that supercomplex and have the special task of channeling electrons effectively through the electron transport chain [59]. Nevertheless, Guo et al. proposed an even larger system called megacomplex that includes complex II at the previous respirasome [42]. They suggested that a quinone/quinol (oxidized/reduced forms of the same molecule) pool maximizes the oxide-reduction reactions. Recent studies suggest that there are around 100 Å between complex I and complex III when actively translocating electrons, proposing with this that there is no need for a mediating protein to help the electron channeling through these complexes [42,60].

MRC-I is the first in the mega-complex that encounters the quinone site to start the oxide-reduction process. This complex is composed of several subunits, and mainly the ND1 subunit is the one that possesses the majority of the quinone binding domain and, to a lesser extent, the ND3, PSST, and 49 kDa subunits. Fiedorczuk et al. studied the open and close positions of the above-mentioned complex I to be active and inactive, respectively [61]. The ND1 subunit has a predominantly structural conformation of alpha-helices that provides the hydrophobic environment expected of a membrane protein and owns the quinone binding domain which is in its core (Table 1).

### 3.1. Structural Analysis

#### 3.1.1. Rotenone and Annocatacin B

Before performing the MD simulations, we carried out QM calculations to obtain the optimized structures and analyze the electrostatic properties of the ligand molecules. Figure 1a shows the 2D representation of the ligands, where we can visualize that annocatacin B is larger than rotenone. The optimized structure of annocatacin B shows a closed isoform between the THF rings and the γ-lactone ring (Figure 1b). This result is in agreement with that observed by Nakanishi et al., who reported that the hydrophobic alkyl tail of the acetogenins, in general, looks to serve as a spacer to accommodate the polar hydroxylated bis-THF motif to the polar-membrane part, and its apolar counterpart, the γ-lactone ring, into the core of the lipid bilayer [62].

Both ligand molecules have an electrophilic character and one of the major goals of this study aimed to assess the electrostatic effect of the ligands on the ND1 structure. Figure 2 shows the quantum and classical ESP surfaces of annocatacin B and rotenone molecules obtained from Hirshfeld population analysis. We can observe that the annocatacin B structure has a high electron density region over the γ-lactone ring and it decreases at the THF rings (Figure 2a). On the other hand, as can be seen from the ESP surface of the rotenone molecule, the high electron density sites are close to the carbonyl group, and the oxygen atoms, as expected (Figure 2b). With these charges, and using the OPLS/AA parameters, we built the annocatacin B and rotenone force fields for the MD simulations. Hirshfeld’s atomic charges calculation and their use in molecular mechanics (MM) force fields has been employed in many liquid solvents studies [63,64,65,66,67]. The main advantages of these atomic charges are not to overestimate the electrostatic properties and accelerate the MD calculations.

Additionally, the drug-like properties of annocatacin B and rotenone have the following values: six hydrogen bond acceptors in both of them; hydrogen bond donors of 2 and 0; molecular weight of 578.875 g/mol and 394.423 g/mol; the number of rotational bonds of 23 and 3; partition coefficient LogP of 8.1069 and 3.7033, and a surface area of 250.531 Å${}^{2}$ and 168.525 Å${}^{2}$, respectively. These results confirmed that both molecules are very hydrophobic, annocatacin B being more lipophilic than rotenone, due mostly to its alkyl chain.

The pharmacokinetic properties of absorption, distribution, metabolism, excretion, and toxicity (ADMET) are in Table 2. The absorption is similar in both compounds; however, rotenone is not a P-glycoprotein substrate giving a slim advantage to the other molecule. The distribution property is slightly higher for rotenone, which implies that its distribution in the human body (tissues) is a bit greater than annocatacin B. Regarding metabolism, both could be substrates of the CYP3A4 protein, but only rotenone could act as an inhibitor. The excretion and toxicity of these molecules are similar in both cases. In general terms, this description shows that both rotenone and annocatacin B have very similar properties. The pharmacokinetic and toxicological properties of these compounds were analyzed through the pkCSM server [68].

#### 3.1.2. ND1—Ligand Complexes and Stability Descriptors

As we said earlier, ND1 subunit is located in the transmembrane region of human MRC-I [42]. In order to understand the ligand effect on its structure, we carried out MD simulations of a full-length ND1 subunit and its ND1-ligand complexes. To obtain the molecular systems, we isolated the ND1 protein of the MRC-I and this was embedded inside a phospholipid bilayer (Figure 3a). As said before, according to Kakutani et al., the ND1 subunit has two regions in the active site, Y127 to F198 (site A) and D199 to K262 (site B), which are involved in the quinone binding domain (Figure 3b) [48]. The authors suggest that natural acetogenins prefer to accommodate more likely in site A and synthetic molecules in site B.

Before studying the structural and energy changes of ND1 protein, it was necessary to assess the stability of the molecular complexes during MD simulations. For this purpose, we calculated and plotted the root mean square deviation (RMSD, for additional information, see Appendix A) of the ND1 subunit for all complexes, with respect to its equilibrated structure. With the best molecular docking results (for additional information, see Appendix A), we carried out 500 ns of MD simulations, and we observed that due to the movement restrictions of the lipid bilayer on the ND1 atoms, there is no significant difference between the protein containing the ligands and the one that does not have them. Specifically, the average RMSD of the last 300 ns of the ND1 without ligands was 0.40±0.04 nm, and the average RMSD of the last 200 ns of ND1 with ligands was 0.48±0.02 nm and 0.44±0.02 nm for the systems ND1-rotenone and ND1-annocatacin B, respectively. At a glance, we can notice that the last 200 ns in the three systems is specially stabilized, that is, within the range of the 0.2 nm (2 Å) of deviation permitted. However, when we analyzed the final MD structures, we observed a structural impact of the ligands in the active site (Figure 4a). In both ND1-ligand complexes, the ND1 subunit shows an open conformation to allow ligand stability (Figure 4b,c). In the case of the rotenone complex, the addition of this ligand caused a structural instability observed in the RMSD of the active site, 0.41±0.11 nm against 0.30±0.06 nm of the annocatacin B complex (Table 3 and Figure 5a).

In the case of the radii of gyration (RG), close values were obtained for the ND1-ligand complexes (∼2.12 nm), due to the stability provided by the lipid membrane (Table 3). Similarly, calculations performed in the active site showed few variations among these zones in the three structures (∼1.90 nm, Figure 5b). However, calculations around the *y*-axis, showed that the most opened structure was the ND1—annocatacin B complex (1.77±0.02 nm), being the ND1—rotenone complex the most compacted structure (1.70±0.04 nm, Figure 5c).

Using both results, the active site RMSDs and *y*-axis RG, we performed a free energy landscape (FEL) analysis to obtain the minimal global energy conformations of the ND1 complexes. The FEL maps showed the impact of ligands on the ND1 structure stabilization. In Figure 5d, we can observe that there is only a single conformation cluster (dashed circles in the 2D maps), which indicates the great stability of the ND1 subunit in the lipid membrane. In the case of the ligand complexes, there are four conformation clusters that indicate the destabilization caused by the ligand molecules. However, the location of these clusters was more close in the ND1—annocatacin B complex (Figure 5e,f). According to the results above, the 3D maps showed a large top area in the ND1—rotenone complex and a less top area in the annocatacin B complex, which suggests a more profound stabilization effect by annocatacin B on the ND1 subunit.

#### 3.1.3. Hydrogen Bond Analysis

To elucidate this apparent contradiction between the results obtained from the RMSD and RG analyses, we performed a hydrogen bonds (HB) analysis. Using the *hbond* tool of Gromacs for the MD simulations, and the Hydrogen bonds plugin of VMD for the global minimum energy structures, we obtained the HB interactions based on a cutoff distance of 0.35 nm and a cutoff angle of 30${}^{\circ }$. Initially, we determined the HB formation of the ND1 subunit, both intra and intermolecular (Figure 6a, Table 3). The results showed a greater intramolecular HB formation in the ND1—annocatacin B complex (∼216) and its active site (∼89), but a decrease in the intermolecular interactions, mainly with the lipid bilayer (∼26 and ∼5 for the active site). On the other hand, the ND1-rotenone complex shows a maximum number of interactions with the solvent molecules (∼356) and the most formation of HB with the DPPC molecules (∼31 and 8 to active site). The same trend was presented in the case of the minimal energy structures (parenthesis results). These results suggest that the annocatacin B increases the intramolecular stability of the ND1 subunit contrary to the rotenone molecule, which increases the intermolecular interactions mainly with the solvent molecules that are the main cause of protein instability.

To clarify these stability behaviors, we carried out HB calculations between the ligand molecules and the system components. In Figure 6b, we can see the HB formations of these ligands and all atoms in the molecular complexes. From a statistical perspective at the last 200 ns of MD simulations, there are more HB formations in the annocatacin B complex (∼0.80) than the rotenone complex (∼0.42). Furthermore, the analysis of the ND1—ligand interactions (Figure 6c), showed almost exclusively ligand interactions by annocatacin B on the ND1 subunit (∼0.15) versus rotenone interactions (∼0.01). The results confirm that the annocatacin B stabilizes, in part, the ND1 structure by polar interactions with its nearby residues. In order to identify the active site residues involved in the stabilization interactions, we calculated the HB occupancies in the MD simulations, and the Figure 6d shows the results obtained. In the case of the ND1—rotenone complex, we can observe the greater occupancy value (9.95%) due to the F223. However, only four residues were involved in the polar interactions (L222, F223, A226, and T229). On the other hand, the ND1—annocatacin B complex showed a major number of polar interactions (14), being W185, F223, M233, and L237 residues that had the highest number of occupancy values. Despite the hydrophobic character of the ligand molecules, our hydrogen bonds analysis showed the importance of polar interactions in the ND1 stabilization.

#### 3.1.4. RMSF and B-Factor Analysis

To evaluate local flexibilities of the ND1 subunit and describe the deviations of residues from the average position due to the ligand effects, we performed the root mean square fluctuation (RMSF) analysis. The main fluctuations of the ND1 protein were observed at the unembedded-loop regions, as expected (Table 3). In particular, high RMSF values were located between L33-G36 residues (ND1—annocatacin B complex, ∼0.84 nm), and A249-S251 residues (ND1—rotenone complex, ∼0.86 nm).

Despite the high stability of the active site, a fluctuation analysis was performed to understand the ligand effect in these region. For this purpose, in addition to RMSF calculations, we analyzed the B-factor, also called thermal factor or Debye–Walle factor [69] and we mapped the values on the active site surfaces.

The stable regions in the MD trajectories were used and the nearest neighbor residues B-factor values are shown in Table 4. The RMSF values of the three systems exhibit similar fluctuation values (see Table 3), showing a high stable behavior. However, when we analyze the fluctuation in site B of the active site, we observed more instability in the rotenone complex, which is reflected in its dispersion value (0.17 ± 0.08 nm) as compared with the ND1 and ND1—annocatacin B values (0.14 ± 0.04 nm and 0.13 ± 0.04 nm, respectively). The highest fluctuations were located between A201-F211, and D248-E253 residues (Figure 7a). In embedded active site regions, the ND1 subunit presents high stability, denoted by the green color of the B-factor surface (Figure 7b). However, this stability is altered by the presence of the ligand molecules, making these regions more flexible. Figure 7c shows the rotenone effects on the neighbor residue fluctuations. The presence of white and red zones on the B-factor surface denotes a flexibility increase, especially, the F223 residue shows a high fluctuation value (144.4 Å). Figure 7d shows a zoom of rotenone and its influence zone on the ND1 protein, calculated at a minor distance of 0.5 nm. The interactions with 23 residues are observed in Table 4. On the other hand, the annocatacin B effects on the B-factor surface shows an increase in the number of residues with high fluctuation, being the L79 (179.7 Å) and M225 (231.7 Å) residues that had the highest fluctuation values (Figure 7e). Hence, the total number of residues interacting with the annocatacin B ligand were 37 (Figure 7f). The results suggest that the annocatacin B stabilizes the ND1 structure by size effect and by interaction with different domains out of the active site.

Finally, we included the solvent accessible surface area (SASA) value, that is an important descriptor of the ligand effects over the structure, in which ND1—rotenone has a higher value (179.29 ± 1.47 nm^2^) than ND1—annocatacin B (177.67 ± 1.53 nm^2^) and ND1 without ligand (174.84 ± 1.37 nm^2^).

#### 3.1.5. MM Electrostatic Potential Surfaces

As mentioned above, ND1—ligand interactions mainly have a hydrophobic character and that is demonstrated by their drug-like properties. However, our results show an electrostatic contribution to structure stabilization. Thus, using the molecular mechanics adaptive Poisson–Boltzmann solver (APBS) approximation [56], we calculated the ESP surfaces of ND1 subunit and their ligand complexes. For this purpose, we used the minimum energy structures and the Hirshfeld’s atomic charges of the ligand molecules obtained in FEL analysis and QM calculations, respectively.

The electrostatic map of the ND1 structure shows a well-defined charged core surrounded by hydrophobic alpha-helices substructures. The core is formed mainly by the active site residues that confer a high electrophilic character to this region (T73-L117 and L266-I273, Figure 8a). We have observed that the binding domain comprises residues out of the active site and the electrostatic properties of these residues are affected by the ligand interactions. Figure 8b shows the drastic variations in the polar properties of the binding domain due to rotenone, increasing the positively charged regions. In addition, the binding domain seems to close, which would explain the more compactness observed in the radii of gyration analysis in this complex.

On the other hand, the electrostatic changes observed by the annocatacin B presence in the binding domain were less dramatic, yet, conserving the electrophilic character in most of its structure (Figure 8c). The main polar variations were located on the A78, S115, I116, L222, F223, N230, M233, and M234 residues, which increased their nucleophilic character. These electrostatic variations suggest that the structural instability observed in the ND1—rotenone complex can be due to structural changes in the active site.

### 3.2. Binding Free Energy

To analyze the energy properties of rotenone and annocatacin B when forming the ND1—ligand complexes, we carried out MM/PBSA calculations based on the last 200 ns of the MD trajectories. In addition, an energy decomposition analysis per residue was performed to highlight the main residues that contribute to the stability of the complexes. As shown in Table 5, the binding free energy (BFE) of the two complexes was energetically favorable, however, the interaction energy of the ND1—annocatacin B complex (−333.18±2.14 kJ/mol) was lower than that of the ND1—rotenone complex (−218.15±1.78 kJ/mol), indicating that the complexation reaction is more spontaneous, which is according to that reported by Murai et al., where they say that the inhibition potency of natural acetogenins is stronger than that of common synthetic inhibitors [22]. Due to the hydrophobic character of the ligand interactions, the main contributions to $\Delta G_{binding}$ energy were the van der Waals (vdW) and nonpolar solvation terms. In both of them, the binding energy was more favorable to the annocatacin B interactions with the ND1 subunit (∼39%). Furthermore, the electrostatic energy term confirms the polar contribution to the stability of the ND1—ligand complexes as seen in the HB analysis, being higher in the annocatacin B complex. These results suggest that annocatacin B has a better stabilization effect on the whole ND1 structure.

As mentioned above, the ligand interactions involve, besides the active site, residues in other regions of the ND1 subunit allowing its structural stability. The large size of the annocatacin B molecule allows a greater number of energetically favorable contacts with these residues compared to those with which the rotenone molecule contacts (Figure 9a). The energy per residue decomposition shows the different contributions to the binding strength in the ND1—ligand complexes (Figure 9b). The highest binding free energy contribution was presented at active site B in the ND1—rotenone complex with the A226 residue (−19.53±0.24 kJ/mol). This complex showed two regions that favored the interactions with rotenone, namely $r_{1}$ (L79-L85) and $r_{2}$ (A221-M234), being $r_{2}$ a zone that involves HB interactions, which would explain its high contributions to the BFE. On the other hand, in the ND1—annocatacin B complex, the greatest BFE contribution was with Val113 (−14.84±0.54 kJ/mol), residue located outside the active site and denoted as $a_{1}$. In addition, three other favorable regions were obtained in this complex, namely $a_{2}$ (M184-T193), $a_{3}$ (A226-L237), and $a_{4}$ (L266-L271), which suggest a better molecular coupling of annocatacin B into the ND1 protein. Table 6 shows the residues that contributed the most to the BFE for both complexes.

The positive energies in the BFE calculations are associated with unfavorable energy interactions between the protein-ligand complexes, Table 7 shows the residues with the highest positive values. In the rotenone complex, these residues were located previous to the active site, namely S115, W118, and S119, being the W118 residue with the highest energy value (8.39±0.16 kJ/mol). On the other hand, in the annocatacin B complex, we observed a greater amount of residues with positive values, being A116 (8.14±0.15 kJ/mol), F223 (5.35±0.19 kJ/mol), and E143 (2.65±0.18 kJ/mol), the residues with the most significant values. These results seem to indicate that the size of annocatacin B could also have a slight destabilizing effect on the ND1 structure, however, this effect is counteracted by the favorable contributions that stabilize it.

Continuing with the BFE analysis, we used the minimum energy structures obtained in the FEL analysis and the 2D ligand-protein interaction diagrams, in order to visualize the moiety interactions of the ligand molecules. For this task, we plotted the BFE values on the ND1 surfaces and analyzed the energy interactions on the ligand structures. In the rotenone complex, we observed that the main residue interactions were located at the dimethoxychromene moiety of rotenone. The greatest favorable interactions were A226 and F223 residues (dark blue color in Figure 10a), being this last one the residue that more interactions showed with rotenone (up to 7 direct interactions with ND1 residues, Figure 10b). Despite the numerous interactions (5), the energy of A82 residue was just −1.04 ± 0.23 kJ/mol. Another important contributions to the BFE were I81, L85, T229, and N230 residues that interacted with the pyranol and the dihydrofuran moieties of the rotenone molecule, respectively. The rotenone’s atom that involved the main energy contributions was the oxygen of the hydroxyl group located at the pyranoloid ring. Residues with the lowest contribution to the BFE were located on the methoxy groups, being S115, W118, and S119 residues with high positive values. The non-polar solvation contribution was the predominant energy term in these residues.

In the case of the annocatacin B complex, the aliphatic chains of this ligand involve the majority of the interactions with its amino acid environment. This includes the interactions with V113 and I116, that were the main contribution and no-contribution residues to the binding energy, respectively (Figure 10c,d). The bis-THF rings only showed one interaction with S188 residue, being an unfavorable one to the total BFE, nevertheless, the energy value was low (1.55±0.54 kJ/mol). The γ-lactone ring involved five interactions, three of them important contributors, namely N230 and M234 (−8.30±0.25 and −9.43±0.24 kJ/mol, respectively). The other two interactions with E143 and S188 residues represented unfavorable energies. It is interesting to note that although the F223 residue showed high fluctuations in both molecular complexes, different energy behavior was observed at the interactions with the ligand molecules. At the rotenone complex, the interaction contributed favorably to the BFE with a significant value (−9.41±0.60 kJ/mol), while at the annocatacin B complex, this contribution was unfavorable (5.35±0.30 kJ/mol).

## 4. Conclusions

QM optimizations, MD simulations, and MM/PBSA analyses were performed to evaluate the molecular and energetic complementarity of the annocatacin B molecule in the ND1 subunit of the human mitochondrial respiratory complex I (MRC-I). We compared these results with those obtained through the analysis of the ND1—rotenone complex, owing to the fact that the rotenone molecule is a powerful inhibitor of this MRC-I. ND1 subunit is a transmembrane protein, thus we used a DPPC lipid bilayer as a membrane model for the entire simulations of the ND1 complexes in this work.

The overall analysis revealed a stabilizing effect of the annocatacin B molecule over the ND1 protein structure. Mainly, this could be due to the size and its capability of folding of annocatacin B, which ultimately allowed it to have more interactions with the ND1 nearby residues. We observe an increase in the formation of hydrogen bonds in the ND1—annocatacin B complex through the whole MD trajectory, which suggests a considerable electrostatic contribution to the stability of the complex. Thus, with respect to the ESP surfaces at the active site, while annocatacin B retained the electrophilic pattern of the native active site of ND1, rotenone largely changed it to a nucleophilic one.

An analysis of MM/PBSA showed that hydrophobic interactions were the main energetic component of the relative binding free energy (BFE), hence the important role of the aliphatic annocatacin B chains in their affinity for the ND1 subunit. Several favorable interactions were observed on these chains in the ND1—annocatacin B complex, including residues outside the active site (V113 and L266-L271), which allowed a 35% better energetic coupling than those observed in the ND1—rotenone complex. Despite the high structural fluctuations of the F223 residue in both complexes, a significant energy interaction was observed, favorable to the BFE (−9.41±0.60 kJ/mol) for the rotenone complex, and unfavorable (5.35±0.30 kJ/mol) for the annocatacin B complex. The reason for this behavior could be that the residue F223 prefers electrostatic to hydrophobic interactions.

Our results suggest that the natural annocatacin B molecule could display better inhibitory capabilities than the rotenone molecule, an issue to be taken into account for future research.

## Figures and Tables

**Figure 1 polymers-13-01840-f001:**
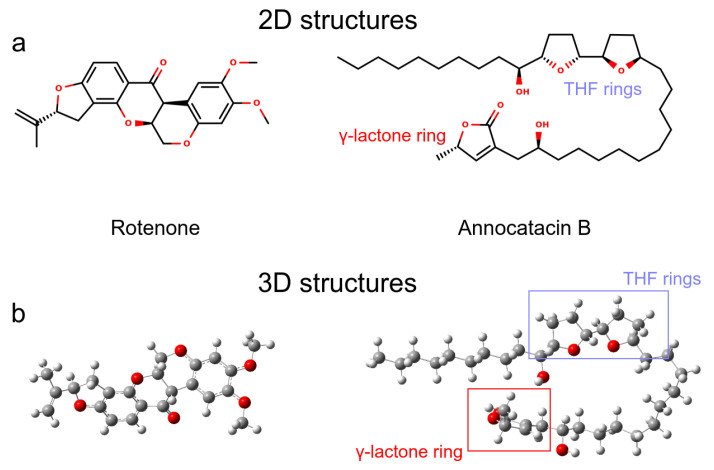
Ligand molecules used in this work. (**a**) 2D representations. (**b**) 3D representations obtained after QM optimization.

**Figure 2 polymers-13-01840-f002:**
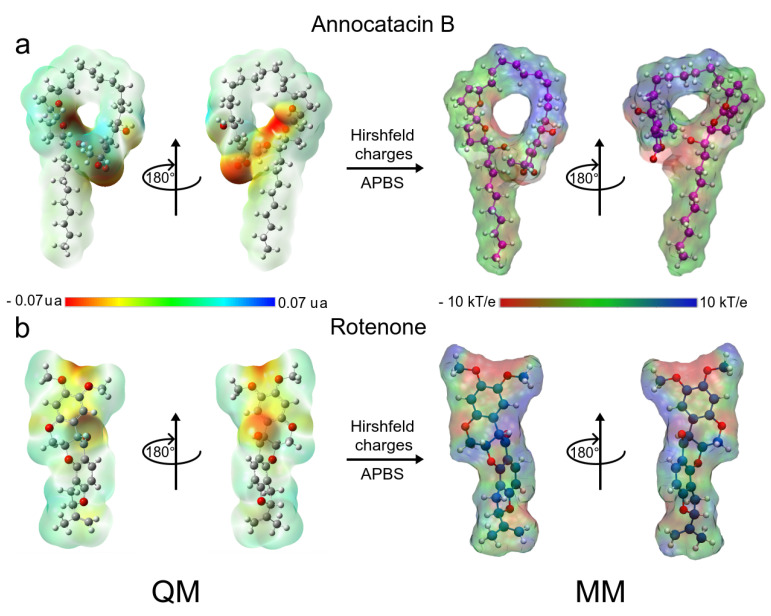
Chemical structure of Annocatacin B and Rotenone molecules. Calculated molecular ESP surfaces of (**a**) annocatacin B, and (**b**) rotenone. In the left panel, ESP surfaces obtained at the DFT level using the CAM-B3LYP/TZVP method. In the right panel, ESP surfaces are obtained with APBS methodology and the Hirshfeld’s atomic charges. On all surfaces, the different colors indicate their molecular electrostatic properties; red for the most nucleophilic zones; dark blue for the most electrophilic zones, and green for the neutral zones.

**Figure 3 polymers-13-01840-f003:**
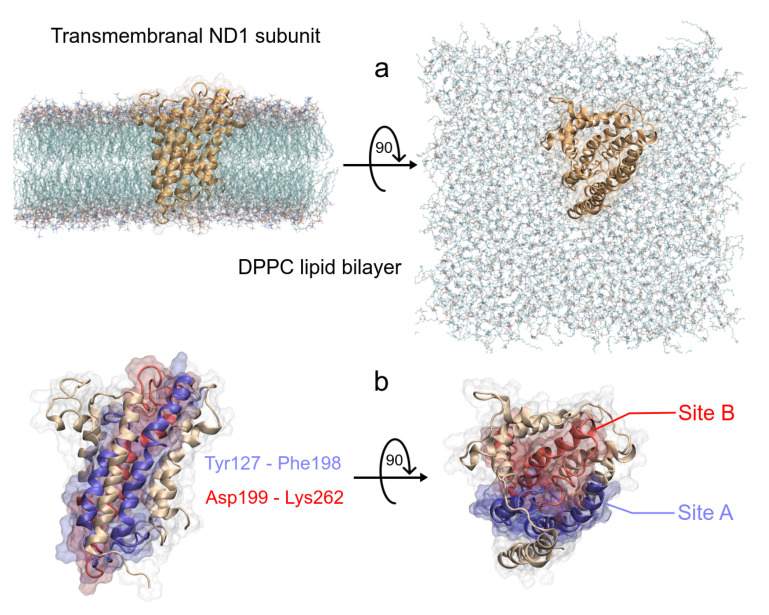
Schematic representation of the ND1—DPPC membrane complex, (**a**) initial distribution of the simulated model used in the MD simulation, the lengths of the simulation box are lx = 11.66 nm, ly = 11.71 nm and lz = 13.02 nm. (**b**) Front and top views of the active site of ND1 protein, in purple color, the active site A, and in red color, the active site B.

**Figure 4 polymers-13-01840-f004:**
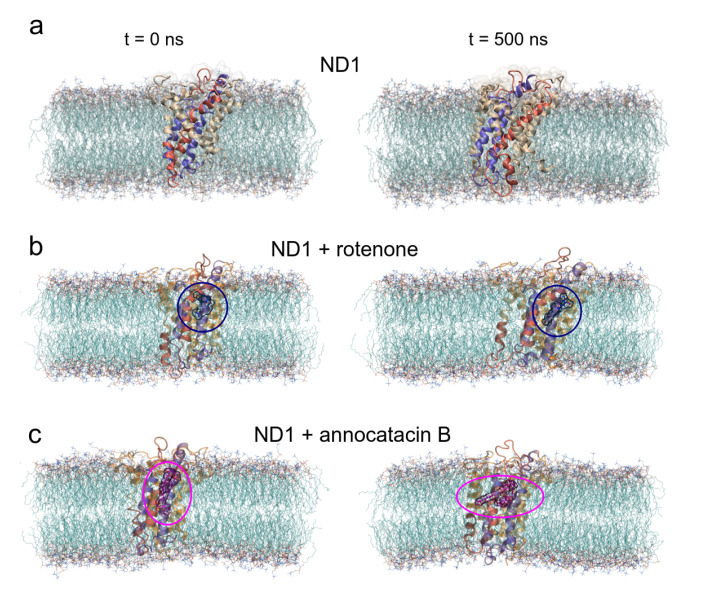
Overall structural organization of ND1 protein and its ligands in a membrane-embedded condition. The left panel shows the molecular complexes at initial conditions (0 ns). The right panel shows the complexes at 500 ns. (**a**) ND1 + DPPC membrane, (**b**) ND1 + DPCC + rotenone, and (**c**) ND1 + DPPC + annocatacin B.

**Figure 5 polymers-13-01840-f005:**
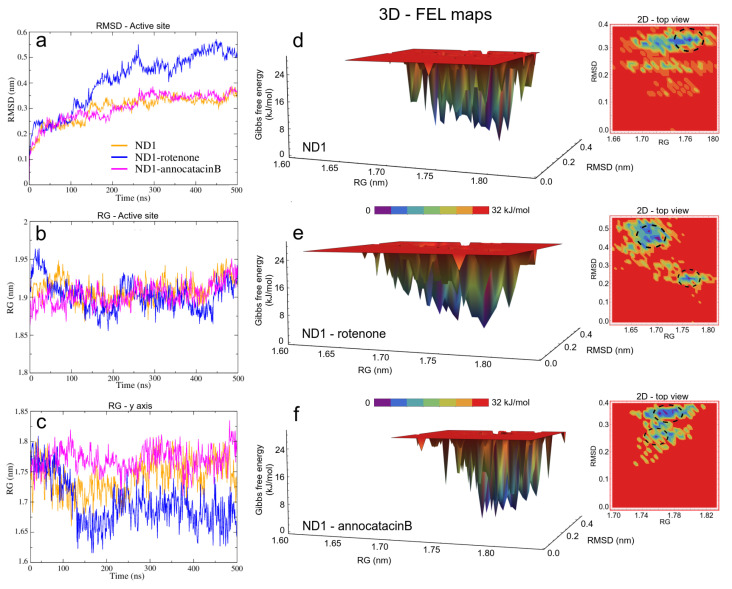
Structural analysis of the ND1 subunit active site. (**a**) RMSD; (**b**) RG; (**c**) RG on of *y*-axis of the ND1 protein. The FEL maps (**d**–**f**) were built using structural coordinates from RMSD results and RG on the *y*-axis. The dash circles in the 2D plots indicate the global minimum energy structures showed in purple color in the 3D plots.

**Figure 6 polymers-13-01840-f006:**
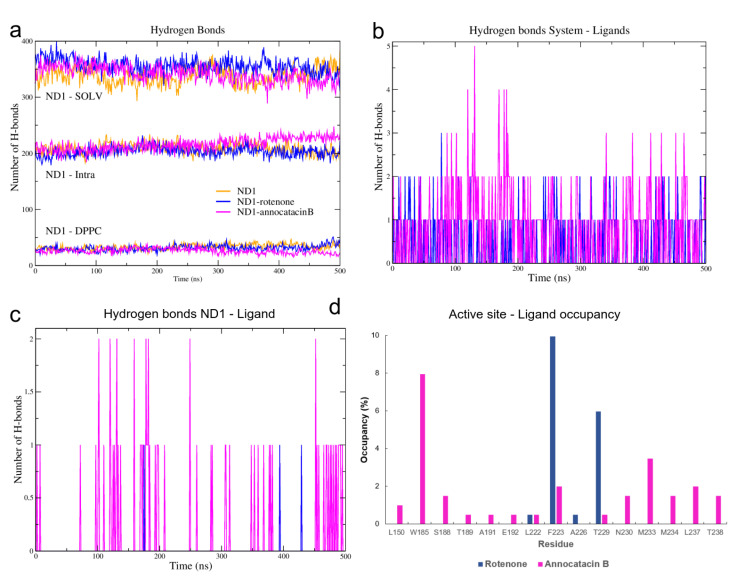
H-bond analysis of the ND1 complexes during 500 ns MD simulations. (**a**) ND1-intramolecular and ND1-intermolecular interactions (solvent and lipid-bilayer). (**b**) H-bond formation between the molecular systems and the ligand molecules (**c**) H-bond formation taking into account just the ND1 subunit and the ligand molecules. (**d**) H-bond occupancy of the active site residues interacting with the ligand molecules.

**Figure 7 polymers-13-01840-f007:**
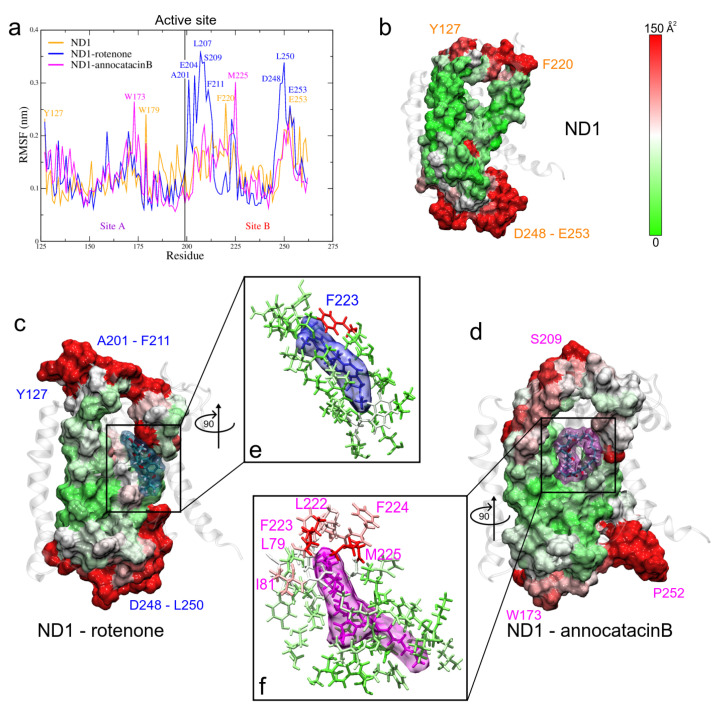
Fluctuation analysis of the ND1 residues. (**a**) RMSF plot of the active site residues obtained during the last 200 ns of the MD trajectories. (**b**–**d**) B-factor plotted on the molecular surface of the active site. The red color indicates high B factor values, whereas the green, low values. Ligands are shown in translucid surface, rotenone in blue color, and annocatacin B in magenta color. (**e**,**f**) Zoom of the ligand interactions.

**Figure 8 polymers-13-01840-f008:**
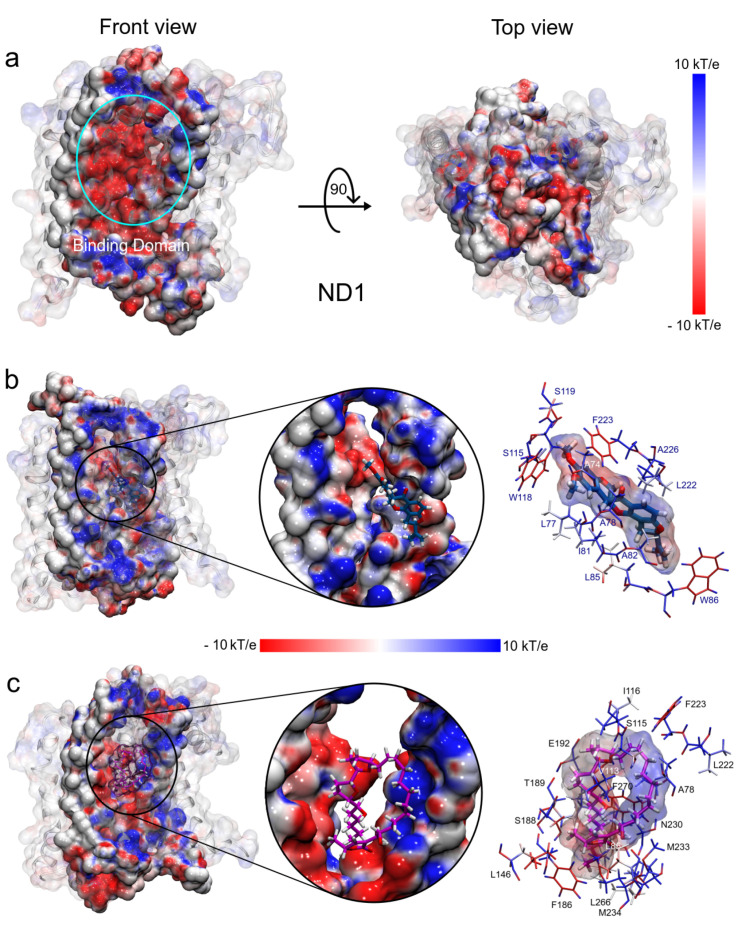
Electrostatic potential surfaces of molecular systems obtained with APBS. (**a**) ND1 protein structure. (**b**) ND1—rotenone complex. (**c**) ND1—annocatacin B complex. The red color indicates negatively charged regions and blue, positively charged. White color denotes hydrophobic regions.

**Figure 9 polymers-13-01840-f009:**
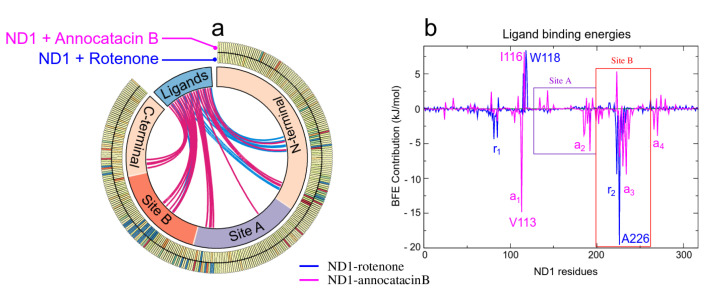
MM/PBSA analysis. (**a**) Circos plot representation of the ND1–ligand structures, where the interactions of the ND1 subunit residues to the ligand molecules are shown as links inthe middle of the plot. The blue lines indicate interactions with rotenone and the magenta lines, with annocatacin B. The outer part of this plot shows the heat map of the BFE per residue, where blue color represents favorable BFE, red color unfavorable BFE, and yellow color indicates neutral energies. (**b**) Energy per-residue contributions plot.

**Figure 10 polymers-13-01840-f010:**
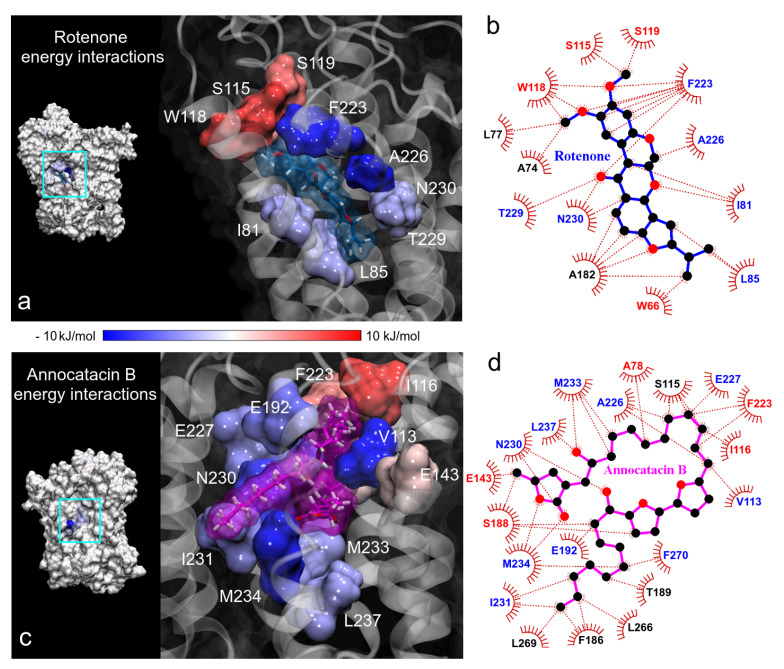
Binding free energy (BFE) plotted on the ND1 surfaces. (**a**,**c**) 3D representation of the main residue contributions to BFE in the ND1 complexes. Blue color indicates favorable energies and red color unfavorable energies. (**b**,**d**) 2D representation of all interactions obtained in contact analysis in the minimum energy structures. The color of residue labels is the same as energy values.

**Table 1 polymers-13-01840-t001:** Secondary structure summary.

System	Strand	Alpha Helix	3–10 Helix	Other	Total Res.
ND1	0 (0.00%)	154 (48.40%)	9 (2.80%)	155 (48.70%)	318
ND1—Annocatacin B	0 (0.00%)	129 (39%)	28 (8.80%)	166 (52.20%)	318
ND1—Rotenone	0 (0.00%)	148 (46.50%)	11 (3.50%)	159 (50.00%)	318

**Table 2 polymers-13-01840-t002:** ADMET prediction of annocatacin B and rotenone by pkCSM server.

ADMET			
**Property**	**Model Name**	**Predicted Value**	
		**Annocatacin B**	**Rotenone**
Absorption	Water solubility ${}^{a}$	−5.85	−5.05
Absorption	Caco2 permeability ${}^{b}$	0.40	1.31
Absorption	Intestinal absorption ${}^{c}$	86.98	99.63
Absorption	Skin Permeability ${}^{d}$	−2.70	−2.75
Absorption	P-glycoprotein substrate	Yes	No
Absorption	P-glycoprotein I inhibitor	Yes	Yes
Absorption	P-glycoprotein II inhibitor	Yes	Yes
Distribution	VDss (human) ^e^	−0.29	−0.04
Distribution	Fraction unbound (human) ${}^{f}$	0.05	0
Distribution	BBB permeability ${}^{g}$	−0.95	−0.87
Distribution	CNS permeability ${}^{h}$	−2.90	−2.82
Metabolism	CYP2D6 substrate	No	No
Metabolism	CYP3A4 substrate	Yes	Yes
Metabolism	CYP1A2 inhibitior	No	Yes
Metabolism	CYP2C19 inhibitior	No	Yes
Metabolism	CYP2C9 inhibitior	No	Yes
Metabolism	CYP2D6 inhibitior	No	No
Metabolism	CYP3A4 inhibitior	No	Yes
Excretion	Total Clearance ${}^{i}$	1.601	0.195
Excretion	Renal OCT2 substrate	No	No
Toxicity	AMES toxicity	No	No
Toxicity	Max. tolerated dose (human) ${}^{j}$	−0.64	0.16
Toxicity	hERG I inhibitor	No	No
Toxicity	hERG II inhibitor	No	No
Toxicity	Oral Rat Acute Toxicity (LD50) ${}^{k}$	3.03	2.87
Toxicity	Oral Rat Chronic Toxicity (LOAEL) ${}^{l}$	0.79	1.43
Toxicity	Hepatotoxicity	No	No
Toxicity	Skin Sensitisation	No	No
Toxicity	T.Pyriformis toxicity ${}^{m}$	0.31	0.35
Toxicity	Minnow toxicity ${}^{n}$	−1.89	−0.33

*^a^* In log mol/L; *^b^* In log Papp in 10^−6^ cm/s; *^c^* In % Absorbed; *^d^* In log Kp; *^e^* In log L/kg; *^f^* In Fu; *^g^* In log BB; *^h^* In log PS; *^i^* In log mL/min/kg; *^j^* In log mg/kg/day; *^k^* In mol/kg; *^l^* In log mg/kg_bw/day; *^m^* In log ug/L; *^n^* In log mM.

**Table 3 polymers-13-01840-t003:** Stability Descriptors of the ND1 complexes.

System	Region	RMSD ${}^{a}$	RMSF ${}^{a}$	RG ${}^{a}$	H B		
					Intra	Inter/Solv	Inter/Mem
ND1	whole prot	0.40 ± 0.02	0.19 ± 0.09	2.12 ± 0.01	209 ± 9 (200)	338 ± 15 (336)	33 ± 5 (39)
	Active site	0.30 ± 0.05	0.13 ± 0.04	1.91 ± 0.01	89 ± 6 (77)	166 ± 10 (165)	6 ± 2 (6)
	*y*-axis			1.74 ± 0.03			
ND1 + rotenone	whole prot	0.48 ± 0.02	0.20 ± 0.10	2.13 ± 0.01	203 ± 8 (202)	356 ± 13 (356)	31 ± 5 (38)
	Active site	0.41 ± 0.11	0.14 ± 0.07	1.90 ± 0.02	84 ± 6 (88)	176 ± 10 (174)	8 ± 3 (10)
	*y*-axis			1.70 ± 0.04			
ND1 + annocatacin B	whole prot	0.44 ± 0.01	0.20 ± 0.09	2.13 ± 0.01	213 ± 11 (207)	341 ± 14 (330)	26 ± 5 (23)
	Active site	0.30 ± 0.03	0.13 ± 0.05	1.90 ± 0.01	89 ± 6 (81)	167 ± 10 (161)	5 ± 3 (2)
	*y*-axis			1.77 ± 0.02			

${}^{a}$ In nanometers. For HB calculations, we considered determining those formed between ND1 subunit itself (intra); the ND1 subunit and solvent molecules (inter/solv); and the ND1 subunit and lipid bilayer membrane (inter/mem). Values between parenthesis were calculated on global minimum energy structures obtained in the FEL analysis. All values were obtained from the last 300 ns of the MD simulations.

**Table 4 polymers-13-01840-t004:** Contact analysis.

System	Active Site				Other Sites		
	Site A		Site B				
ND1 + rot + mem					T73(21)	A74(19)	L77(37)
			L22(23)	F223(144)	A78(15)	L79(45)	I81(20)
			A226(14)	T229(17)	A82(18)	L83(37)	L85(19)
			N230(36)	I232(10)	W86(68)	L89(30)	M91(23)
			M233(20)		S115(23)	I116(38)	W118(41)
					S119(37)		
ND1 + ann + mem			L222(94)	F223(103)	A78(102)	L79(179)	I81(101)
	E143(39)	L146(27)	F224(106)	M225(232)	A82(76)	L85(43)	S109(48)
	W185(14)	F186(41)	A226(70)	E227(48)	A112(40)	V113(53)	Y114(52)
	S188(13)	T189(14)	T229(27)	N230(25)	S115(39)	I116(70)	L117(74)
	A191(14)	E192(15)	I231(37)	M233(49)	L266(28)	T267(19)	L269(46)
			M234(36)	L237(50)	F270(26)	I273(49)	

Residues close to ligand at distance minor to 0.5 nm obtained at the last 200 ns of the MD trajectories. In parenthesis, B-factor values.

**Table 5 polymers-13-01840-t005:** Average MM/PBSA free energies of ND1 complexes.

System	$\Delta E_{\mathrm{VW}}^{a}$	$\Delta E_{\mathrm{Elec}}^{b}$	$\Delta E_{\mathrm{PS}}^{c}$	$\Delta E_{\mathrm{SASA}}^{d}$	${\mathrm{BFE}}^{e}$
ND1-Annocatacin B	−358.76 ± 1.26	−23.04 ± 0.69	85.00 ± 0.92	−36.38 ± 0.10	−333.18 ± 2.14
ND1-Rotenone	−219.81 ± 0.89	−21.14 ± 0.39	45.01 ± 0.34	−22.21 ± 0.06	−218.15 ± 1.78

*^a^* Van der Waals energy.; *^b^* Electrostatic Energy, *^c^* Polar Solvation Energy; *^d^* SASA Energy; *^e^* Binding Free Energy. All values are in kJ·mol^−1^.

**Table 6 polymers-13-01840-t006:** Top 10 residues that does contribute to the binding free energy.

ND1—Rotetone	$\Delta E_{\mathrm{MM}}$	$\Delta E_{\mathrm{PS}}$	$\Delta E_{\mathrm{AS}}$	BFE
A226	−8.18 ± 0.12	0.92 ± 0.06	−12.27 ± 0.19	−19.53 ± 0.24
F223	−14.37 ± 0.20	3.94 ± 0.08	1.06 ± 0.63	−9.41 ± 0.60
I81	−9.21 ± 0.11	0.66 ± 0.02	4.19 ± 0.14	−4.36 ± 0.19
T229	−4.32 ± 0.11	5.20 ± 0.13	−5.06 ± 0.32	−4.17 ± 0.35
L85	−5.97 ± 0.11	0.72 ± 0.07	1.10 ± 0.13	−4.15 ± 0.16
N230	−1.35 ± 0.06	0.72 ± 0.06	−2.86 ± 0.14	−3.49 ± 0.16
L222	−9.03 ± 0.10	2.46 ± 0.07	3.79 ± 0.18	−2.77 ± 0.22
M225	−3.74 ± 0.13	1.19 ± 0.07	0.24 ± 0.12	−2.31 ± 0.13
M233	−2.77 ± 0.07	2.23 ± 0.07	−1.51 ± 0.11	−2.05 ± 0.13
E192	−1.61 ± 0.08	−0.17 ± 0.15	−0.22 ± 0.01	−2.00 ± 0.16
**ND1—Annocatacin B**				
V113	−12.39 ± 0.22	2.74 ± 0.12	−5.20 ± 0.34	−14.84 ± 0.54
M234	−10.30 ± 0.16	3.67 ± 0.08	−2.80 ± 0.16	−9.43 ± 0.24
N230	−10.59 ± 0.19	6.10 ± 0.13	−3.82 ± 0.17	−8.30 ± 0.25
A226	−4.94 ± 0.22	0.60 ± 0.08	−2.59 ± 0.46	−6.92 ± 0.61
E192	−8.48 ± 0.20	1.29 ± 0.35	1.11 ± 0.22	−6.09 ± 0.33
I231	−4.26 ± 0.10	−0.19 ± 0.02	−0.74 ± 0.09	−5.19 ± 0.13
E227	−0.98 ± 0.26	−3.77 ± 0.29	−0.35 ± 0.10	−5.09 ± 0.30
M233	9.20 ± 0.16	4.13 ± 0.08	0.61 ± 0.09	−4.46 ± 0.18
L237	−6.74 ± 0.20	1.18 ± 0.03	1.15 ± 0.07	−4.40 ± 0.19
F270	−7.34 ± 0.16	1.39 ± 0.06	2.06 ± 0.10	−3.90 ± 0.15

All values are in kJ·mol^−1^.

**Table 7 polymers-13-01840-t007:** Top 10 residues that does not contribute to the binding free energy.

ND1—Rotetone	$\Delta E_{\mathrm{MM}}$	$\Delta E_{\mathrm{PS}}$	$\Delta E_{\mathrm{AS}}$	BFE
W118	−7.31 ± 0.12	2.65 ± 0.07	13.05 ± 0.30	8.39 ± 0.30
S119	−1.11 ± 0.04	1.12 ± 0.07	6.11 ± 0.23	6.12 ± 0.22
S115	−4.37 ± 0.11	1.67 ± 0.06	7.78 ± 0.20	5.09 ± 0.20
W86	−4.52 ± 0.10	0.93 ± 0.05	5.21 ± 0.25	1.61 ± 0.25
K262	1.14 ± 0.01	−0.13 ± 0.01	0.22 ± 0.00	1.23 ± 0.01
R195	1.61 ± 0.03	−0.58 ± 0.04	0.00 ± 0.00	1.03 ± 0.05
E214	1.53 ± 0.03	−0.75 ± 0.02	0.07 ± 0.00	0.85 ± 0.02
Y215	−0.47 ± 0.02	0.41 ± 0.03	0.91 ± 0.12	0.84 ± 0.13
A78	−9.30 ± 0.11	2.64 ± 0.06	7.37 ± 0.23	0.71 ± 0.29
E59	1.05 ± 0.02	−0.30 ± 0.01	−0.17 ± 0.00	0.58 ± 0.01
**ND1—Annocatacin B**				
I116	−3.78 ± 0.09	0.51 ± 0.03	11.40 ± 0.24	8.14 ± 0.23
F223	−4.83 ± 0.13	2.67 ± 0.15	7.50 ± 0.25	5.35 ± 0.30
E143	−7.08 ± 0.23	8.98 ± 0.56	0.70 ± 0.10	2.65 ± 0.52
A78	−3.20 ± 0.08	2.23 ± 0.09	3.42 ± 0.15	2.44 ± 0.16
R274	−1.97 ± 0.13	4.04 ± 0.10	−0.08 ± 0.01	1.98 ± 0.13
S109	−5.27 ± 0.17	5.55 ± 0.15	1.55 ± 0.10	1.82 ± 0.21
R134	1.12 ± 0.02	0.65 ± 0.03	−0.16 ± 0.00	1.61 ± 0.04
R195	0.83 ± 0.04	0.74 ± 0.03	0.01 ± 0.00	1.59 ± 0.04
R34	−0.24 ± 0.06	1.88 ± 0.06	−0.08 ± 0.00	1.55 ± 0.08
S188	−6.55 ± 0.12	6.38 ± 0.14	1.72 ± 0.09	1.55 ± 0.19

All values are in kJ·mol^−1^.

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
