# Peer review of "Structural and Energetic Affinity of Annocatacin B with ND1 Subunit of the Human Mitochondrial Respiratory Complex I as a Potential Inhibitor: An In Silico Comparison Study with the Known Inhibitor Rotenone"

_polymers, 2021, doi:10.3390/polym13111840_

Round 1

Reviewer 1 Report

The ms deals with all-atom molecular dynamics (MD) simulations of a subunit of the mitochondrial respiratory complex I embedded in a slab of a lipidic bilayer. The authors compare its stability when coupled with a couple of ligands, rotenone and annocatacin B. For building the protein model, they clean the PDB structure from water molecules and set up the OPLS-AA force field. For the case of the ligands, they also implement the OPLS-AA force field and carried out a Hirshfeld in-vacuo calculation to set partial charges on the sites. Finally, they performed a docking analysis to produce a reasonable initial configuration of both protein-ligand systems. Then, MD simulations were run during 500 ns and the trajectory was analyzed through standard techniques. The authors conclude that annocatacin B is more effective to stabilize the protein than rotenone. In my opinion, this is a questionable conclusion. In addition, the authors should provide more details to make it easier to reproduce their study. Finally, the writing of the ms must be improved. Thus, I cannot recommend its publication in the present form. The following is a list of issues the authors must take into account. 

Main issues

1- 500 ns is well below the relaxation time of such big molecules. Thus, results are strongly dependent on the selected initial configuration. This makes it extremely important to take into account different initial configurations close to the free energy minimum. You are only considering one. How different is the docking of the best five configurations? How can you be sure the discarded configurations are not relevant to the real free energy? At least, you should warn the reader of this limitation. 

2- The Hirshfeld in-vacuo calculation of partial charges may not provide charges compatible with the OPLS-AA force field. This is simply because the OPLS-AA force field is empiric and somehow accounts for the solvent molecules surrounding the solute. You should also warn the reader about this issue. 

3- The water model of the OPLS-AA force field is the TIP. Why did you use the SPC water model instead?

4- Molecular Mechanics Poisson-Boltzmann Surface Area calculations have their strong limitations. For instance, they do not take into account the conformational entropy contribution and are based on a mean-field approximation for the distribution of ions. Again, you cannot blindly take their outcomes as the pure truth or try to mislead your readers on their validity.             

Given all these comments, you must revisit your conclusions stressing all assumptions made during the analysis. 

5- Provide the necessary files to produce the trajectories using gromacs. This is to make your study more transparent and allow others to reproduce your data.  

6- Title: You should stress the comparison between the rotenone and the annocatacin B as inhibitors in it instead of the tools you are using. Also, avoid using acronyms in the title and abstract. 

Minor points

Acronyms must be defined the first time you use them. The point is making the text easy to read. 

Units must multiply both, the magnitude and the error. For instance, (23 +/- 5) N is correct but  (23 +/- 5 N) is not. 

It is a good practice to only employ a single digit to express errors. Sometimes, when the first digit is one, then some people advise employing two digits. Nonetheless, the consensus is to never use more than two. There is a statistical basis for this. For instance, if you have doubts about the units of 179.29 nm 2, reporting the cents has no sense at all, and so the last 9 is meaningless.     

It says: Some hypotheses and studies show that, to a greater or lesser extent, neoplastic cells have many phenotypes related to their energy production, from high aerobic glycolysis, through a partially active oxidative phosphorylation, to a highly productive one [6,7]. Although this issue is controversial [8]. - > Although controversial [], some hypotheses and studies show that ...

The main inhibitor of the MRC-I is the rotenone molecule - > The main known inhibitor? You are concluding it is not. 

... it is highly neurotoxic due to its lipophilic nature - > being lipophilic does not imply neurotoxicity. Please, rephrase. 

containing oxygen-containing -> avoid this. 

to generates them - > fix

The InflateGRO methodology - > Avoid this. Instead, explain what the InflateGRO does. 

were removed in Chimera UCSF 1.11.2 - > Explain what it does.

water-model molecules and ions were added to neutralize the systems. - > water-model molecules do not neutralize the systems. 

The equilibrium MD simulation was realized with position restraint in two
ensembles. - > You must indicate what atoms are restraint. 

Periodic boundary conditions (PBC) in all directions, Particle Mesh
Ewald (PME) algorithm for long-range electrostatics with cubic interpolation with a cut-off of 0.9 nm, and Linear Constraint Solver (LINCS) with all bonds constrained were applied for all MD simulations. - > These conditions appear at the end of the description but are in common for all simulations. You should place this before explaining the different simulations you are doing. 

... were in agreement with the experimental data. -> A reference is needed. 

Based on the g_mmpbsa program [49]. - > Avoid this. You are assuming all potential readers know what the  g_mmpbsa program does.  

partition coefficient LogP of 8.1069 and 3.7033, and a surface area of
250.531 Å 2 and 168.525 Å 2. What are the errors for each determination?

ADMET prediction of annocatacin B and rotenone by pkCSM server. - > Please, include errors to these numbers. 

Schematic representation of the ND1 - DPPC membrane complex - > Schematic representation sounds like a drawing. I understood this image is obtained from simulations. It is better to say it is a snapshot. 

The authors suggest that natural acetogenins prefer to accommodate more likely in site A and synthetic molecules in site B. Why?

The RMSF plot of the three systems shows that the ND1 - rotenone complex had the highest fluctuation (0.14 ± 0.06 nm), compared
with the ND1 (0.13 ± 0.04 nm) and ND1 - annocatacin B (0.13 ± 0.05 nm) complexes (Figure 7a). You cannot say (0.14 ± 0.06) nm is higher than (0.13 ± 0.04) nm. Both values lie in-between the error bars of the other. They are statistically the same. 

Figure 8b shows the drastic variations in the polar properties of the binding domain due to rotenone, increasing the positively charged regions. What increases, the surface of the positively charged regions or the charge of the positively charged regions? 

negative polar character  - > There is a polar character, negative charge, and positive charge, but there is not a negative polar character. 

The main polar variations were located on the A78, S115, I116, L222, F223, N230, M233, and M234 residues, which increased their positive charge - > if so, since the charge is a fixed property of the model molecule, there should appear other residues compensating this charge shift. 

the interaction energy of the ND1 - annocatacin B complex was more spontaneous  - > The interaction free energy is a number. A number cannot be more spontaneous. The number just indicates that the complexation reaction is more spontaneous. 

was the hydrophobic interactions. - > fix.

Author Response

We appreciate the reviewer's comments, and we respond point by point in the attached pdf document.

Reviewer 2 Report

The work by Molina et al is well presented and all simulations are well performed. They prefer to run on single long MD of 500ns instead to perform i.e. 3 replica of short MD or similar to improve the sampling. Could the authors comment on that? This is typical procedure in MMPBSA calculation. How they estimated the convergence of the binding free energy? This is an important task during MMPBSA. Please add the options used in the PB calcualtions.

Author Response

(The authors gave the same response as above.)
